# Aldo-Keto Reductase 1C1 (*AKR1C1*) as the First Mutated Gene in a Family with Nonsyndromic Primary Lipedema

**DOI:** 10.3390/ijms21176264

**Published:** 2020-08-29

**Authors:** Sandro Michelini, Pietro Chiurazzi, Valerio Marino, Daniele Dell’Orco, Elena Manara, Mirko Baglivo, Alessandro Fiorentino, Paolo Enrico Maltese, Michele Pinelli, Karen Louise Herbst, Astrit Dautaj, Matteo Bertelli

**Affiliations:** 1Dipartimento di Riabilitazione, Ospedale San Giovanni Battista, A.C.I.S.M.O.M., 00148 Rome, Italy; s.michelini@acismom.it (S.M.); a.fiorentino@acismom.it (A.F.); 2Istituto di Medicina Genomica, Università Cattolica del Sacro Cuore, 00168 Rome, Italy; pietro.chiurazzi@unicatt.it; 3Fondazione Policlinico Universitario “A.Gemelli” IRCCS, UOC Genetica Medica, 00168 Rome, Italy; 4Dipartimento di Neuroscienze, Biomedicina e Movimento, Sezione di Chimica Biologica, Università di Verona, 37134 Verona, Italy; valerio.marino@univr.it (V.M.); daniele.dellorco@univr.it (D.D.); 5MAGI Euregio, 39100 Bolzano, Italy; elena.manara@assomagi.org (E.M.); mirko.baglivo@assomagi.org (M.B.); 6MAGI’s LAB, 38068 Rovereto, Italy; paolo.maltese@assomagi.org; 7Dipartimento di Scienze Mediche Traslazionali, Sezione di Pediatria, Università di Napoli Federico II, 80131 Naples, Italy; michele.pinelli@unina.it; 8Telethon Institute of Genetics and Medicine (TIGEM), 80078 Pozzuoli, Italy; 9Departments of Medicine, Pharmacy, Medical Imaging, Division of Endocrinology, University of Arizona, Tucson, AZ 85721, USA; karenherbst@email.arizona.edu; 10EBTNA-Lab, 38068 Rovereto, Italy; astrit.dautaj@assomagi.org

**Keywords:** lipedema, subcutaneous fat, *AKR1C1*, aldo-keto reductase activity, steroid hormone metabolism, whole exome sequencing, molecular modelling

## Abstract

Lipedema is an often underdiagnosed chronic disorder that affects subcutaneous adipose tissue almost exclusively in women, which leads to disproportionate fat accumulation in the lower and upper body extremities. Common comorbidities include anxiety, depression, and pain. The correlation between mood disorder and subcutaneous fat deposition suggests the involvement of steroids metabolism and neurohormones signaling, however no clear association has been established so far. In this study, we report on a family with three patients affected by sex-limited autosomal dominant nonsyndromic lipedema. They had been screened by whole exome sequencing (WES) which led to the discovery of a missense variant p.(Leu213Gln) in *AKR1C1*, the gene encoding for an aldo-keto reductase catalyzing the reduction of progesterone to its inactive form, 20-α-hydroxyprogesterone. Comparative molecular dynamics simulations of the wild-type vs. variant enzyme, corroborated by a thorough structural and functional bioinformatic analysis, suggest a partial loss-of-function of the variant. This would result in a slower and less efficient reduction of progesterone to hydroxyprogesterone and an increased subcutaneous fat deposition in variant carriers. Overall, our results suggest that *AKR1C1* is the first candidate gene associated with nonsyndromic lipedema.

## 1. Introduction

Lipedema is a chronic debilitating disorder affecting subcutaneous (SC) adipose tissue characterized by bilateral increased circumference of extremities, pain sensations, and bruising. It leads to a disproportionate body shape [1] with subcutaneous fat accumulation in the lower and also upper extremities that can result in considerable disability. The hypertrophic fat pads are typically unresponsive to dietary regimens or physical activities [2]. The disease develops almost exclusively in females during or after puberty, pregnancy, or even menopause, and often in conjunction with sexual hormonal changes. Moreover, anxiety and depression constitute important psychological comorbidity in women with lipedema [1,3]. Although the condition is well described, and an estimated 8 to 17% of adult women worldwide are affected, it is still often underdiagnosed.

Self-reported positive family history of lipedema has been found for up to 64% of women, therefore, a genetic etiology for lipedema is strongly suggested [4]. This disease can be differentiated in nonsyndromic and syndromic forms. Eleven genes involved in seven different comorbidities related to syndromic lipedema have already been identified [5]. The causes of nonsyndromic lipedema are unclear and no genetic components have been identified yet, however familial cases of nonsyndromic lipedema are common, and a genetic cause has been suspected. A study from 2010 showed that within six families of more than three generations with lipedema, a genetic autosomal-dominant hereditary pattern with sex limitation was found [6].

Lipedema is an almost exclusive sex-restricted disease of women. Sex differences include a larger subcutaneous (SC) adipose tissue in women as compared with men, and the relationship of this tissue with the production of steroid hormones has been well described [7]. It is known that sex hormones also determine the anatomical site of the accumulation of adipose tissue [8,9]. Western blot analysis has demonstrated that both progesterone receptor isoforms (PR-A and PR-B) were present in human SC adipose tissue [10]. Steroid production levels influence each other and a change in their metabolism leads to several consequences on subcutaneous fat. For example, estradiol is important to mobilize adipose energetic reserve, and in the brain, it contributes to the regulation of body energy homeostasis [3,11], whereas, in rats, progesterone reverses the weight-reducing actions of estradiol [12,13]. This suggests a differential type of regulation of the SC adipose tissue cells by different sex steroids.

Growing evidence suggests that sex hormone-specific effects could be one of the key biological features for higher mood disease prevalence in women [14]. Neurosteroid hormones or their derivatives influence the regulation of the anxiety-related brain functions, thereby modulating individual anxiety states [14]. For example, derivatives of progesterone, pregnanolone, and allopregnanolone have been shown to be highly selective and potent allosteric modulators of GABA_A_ receptors playing a pivotal role in anxiety [15]. Moreover, abnormal neurosteroidogenesis is implicated in pathological conditions associated with a dysregulation of neuronal inhibition, such as pathological anxiety and depression [16]. Neurosteroids are also important in the regulation of pain perception. Some studies have highlighted that progesterone and its derivatives, dihydroprogesterone and allopregnanolone, had a specific neuroprotective action in the central and peripheral nervous system [17]. Allopregnanolone has also been proven to exert an analgesic effect in various pain models such as the sciatic nerve crush injury model [18].

In summary, dysfunction of sex steroids results in abnormal fat distribution in predisposed subjects, especially in females at the time of puberty [7]. The homeostasis of steroid hormones is finely regulated by hydroxysteroid dehydrogenase (HSD) enzymes expressed in adipocytes that constitute SC adipose tissue. In isolated mature adipocytes, progesterone is converted to 20-hydroxyprogesterone as the main metabolite, most likely through the activity of aldo-keto reductases, a class of HSDs. In particular, AKR1C1 (aldo-keto reductase family 1 member C1) predominantly inactivates progesterone into 20-α-hydroxyprogesterone via its 20α-HSD activity, indirectly regulating the adiposity of SC fat [3,11].

In this work, we analyzed a family with apparent monogenic nonsyndromic lipedema in order to find the causative gene. We employed whole exome sequencing (WES) and identified a genetic variant of *AKR1C1* whose effect has been thoroughly investigated in silico with bioinformatic tools [19], which suggested a partial loss-of-function (LoF) associated with the identified variant.

## 2. Results

### 2.1. Identification of a Missense AKR1C1 Variant in Lipedema Patients

Given the absence of known genes associated with nonsyndromic lipedema and given the apparently sex-limited autosomal dominant transmission of the condition in this family, we performed a WES analysis to identify the responsible variant. This rare pedigree is composed of 12 individuals, three affected and nine healthy (Figure 1). The analysis was performed by sequencing exons and intron-exon junctions of all known genes and focused on heterozygous variants, present in the affected patients (black circles) and absent in all control family members (white circles and squares in Figure 1). Subsequently, these variants were filtered by removing those present in the other 22 unrelated individuals sequenced by WES and those present in more than 1% of the control subjects from the gnomAD population database (https://gnomad.broadinstitute.org/). Finally, all synonymous variants were removed from the list. Variants that segregated with the affected phenotype are reported in Appendix A.

Although none of the variants was found in genes already associated with syndromes that include lipedema, some variants reside in genes that regulate steroid hormone signaling and are involved in causing abdominal obesity or metabolic syndrome. These variants can cause lipedema directly or contribute to its multifactorial etiology. For example, one variant has been found in the *NGEF* gene, which has been associated with abdominal obesity [20,21], while another has been found in *FBXL7*, which has been associated with metabolic syndrome [22] and with an altered pharmacological response to corticosteroids [23]. However, the most promising variant (c.638T > A; p.Leu213Gln) was found in *AKR1C1*, a gene that has been involved in progesterone metabolism [24] and is highly expressed in adipocytes and subcutaneous fat. Sanger sequencing confirmed the presence of this missense variant (indicated in short as L213Q) in the three affected females and excluded it in the unaffected family members (data not shown). Then, we performed real-time qPCR on total RNA extracted from blood of family members to evaluate the effect of the variant *AKR1C1* mRNA stability and, as shown in Figure 2, we found no difference between affected (black bars) and unaffected (white bars) family members.

### 2.2. Structural Analysis and Molecular Dynamics Simulations

The three-dimensional structure of AKR1C1 shows the typical architecture of an alpha-beta barrel, specifically exhibiting the triose-phosphate isomerase (TIM) barrel fold, consisting of eight β-strands coupled with their respective α-helix (Figure 3A). Residue Leu213 was located in the “core” region of the protein (Figure 3A), on the outward side of seventh β-strand of the β-barrel constituting the so-called “pore” region, and it was found to be involved in a network of highly persistent hydrophobic interactions (Figure 3B) with residues L191 (81.7% persistence over the 1 µs Molecular Dynamics (MD) simulations), V265 (94.6%), L202 (27.1%), and L203 (96.4%). The L213Q variant associated with lipedema was unable to create these hydrophobic interactions due to the physicochemical nature of its polar sidechain, resulting in a destabilization of the hydrophobic network surrounding residue 213. Indeed, the persistence of the interaction between V191 and L202 decreased from 22.2% in the WILD-TYPE to 17.9% in the L213Q variant, similar to the behavior exhibited by the interaction between V265 and L203, whose persistence decreased from 63% to 60.9%. In addition, the presence of a polar sidechain gave rise to the formation of novel, yet less persistent hydrogen bonds (Figure 3C) with N189 (13.8%), C193 (43.4%), Q199 (53.9%), C206 (19.9%), and R263 (18%).

Despite the rearrangement of the interaction network of residue 213, MD simulations suggest that the lipedema-associated variant does not dramatically affect the three-dimensional structure of AKR1C1. Indeed, as shown by the RMSF profiles (Appendix A), the two variants exhibited a very similar protein flexibility, with L213Q variant showing a small reduction in the regions encompassing residues 68–80 and 165–180. Interestingly, the largest differences occurred in two out of the three loops representing the steroid-binding cavity, namely the C-terminal part of loop C (310–320), where the L213Q variant seemed to be less flexible, and loop A (117–134), where the WILD-TYPE was significantly more structurally stable, suggesting an allosteric effect of the variant.

To evaluate the effect of the structural rearrangement of the steroid-binding loop, we monitored the solvent accessibility of the hPGS and NADP+, as well as the interaction energy between protein, steroid, and cofactor. The results, summarized in Table 1, suggest that both hPGS and NADP+ are significantly more solvent-exposed in the case of the L213Q variant (Appendix A), specifically the solvent-accessible surface of NADP+ increased from 1.10 ± 0.26 to 1.55 ± 0.35 nm^2^, whereas that of hPGS increased from 0.98 ± 0.43 to 1.46 ± 0.59 nm^2^. Such an increase in solvent accessibility resulted in a substantial decrease of the interaction energy between hPGS and both AKR1C1 (−122.91 ± 23.60 kJ/mol vs. −105.66 ± 23.88 kJ/mol) and NADP+ (−9.75 ± 7.74 kJ/mol vs. −5.22 ± 6.17 kJ/mol), thus, implying a loss of non-covalent interactions between protein, substrate, and cofactor.

### 2.3. QSAR Models Predict a Partial Loss of Function for The L12Q AK1RC1 Variant

A quantitative structure–activity relationship (QSAR) model was built to predict the enzymatic parameters (the turnover number *k_cat_* and the enzyme catalytic efficiency *k_cat_*/*K_m_*) described in [25] for the AKR1C1 variant using the structural and energetic descriptors derived from 20 ns MD simulations of WILD-TYPE AKR1C1 and its variants. Such methods have been proven to effectively predict the functional effects of variants on different classes of enzymes such as serine proteases [26].

The descriptors reported in Table 1, whose differences were proven to be statistically significant (one-tailed *t*-test, *p* < 0.001), were tested for single linear correlation with the log_10_ of the three functional descriptors. The highest single correlation value (*R*^2^ = 0.72, Appendix A) was obtained by the combination (model A) of log *k_cat_* and the interaction energy between AKR1C1 and hPGS (*IE_P-hP_*), resulting in the following model:(1)$$\text{model}\;A:\;\log k_{cat}=-3.521-\left(0.0444*IE_{P-hP}\right)$$

Noticeably, although the combination log *k_cat_* and SAS hPGS also showed a satisfactory *R*^2^ value (0.68), the two independent variables (*IE_P-hP_* and SAS hPGS) resulted to be correlated (*R*^2^ = 0.63, Appendix A), therefore such model was discarded. In addition, the double linear correlation with the combination *IE_P-hP_* and *IE_N-hP_* resulted in a high *R*^2^ (0.77, Appendix A) but the correlation was found to be substantially dependent on *IE_P-hP_*, leading to the rejection of the model due to redundancy.

No single linear correlation was found between any of the four MD descriptors and any of the other two functional parameters log(*K_m_*) and log(*k_cat_*/*K_m_*) (Appendix A). Then, a double linear correlation model was employed for SAS-based and IE-based descriptors after assessing the independence of each couple of variables using single linear regression (*R*^2^ = 0.32 and 0.13, respectively, Appendix A). No double linear correlation was found again with respect to the log*K_m_*, while the best double linear correlation (*R*^2^ = 0.73, Appendix A) was obtained by the model combining the *IE_P-hP_* and the interaction energy between hPGS and NADP+ (*IE_N-hP_*) with the log*k_cat_*/*K_m_* (model B) as follows:(2)$$\text{model}\;B:\;\log\frac{k_{cat}}{K_{m}}\;=\;-7.581-\left(0.0769*IE_{P-hP}\right)+\left(0.0426*IE_{N-hP}\right)$$

Both models were validated using the leave-one-out cross validation method, resulting in an R^2^ = 0.70 ± 0.16 for model A and R^2^ = 0.74 ± 0.06 for model B, with MSE = 0.125 and 0.394, respectively. Then, the QSAR models were used to predict the effects of the L213Q substitution on AKR1C1 catalytic activity.

Interestingly, the estimation of the Michaelis constant (*K_m_*), the catalytic constant (*k_cat_*), and the catalytic efficiency (*k_cat_*/*K_m_*) resulted in a 20% increase in *K_m_* associated with a 41% reduction of the *k_cat_* and an almost halved catalytic efficiency (Table 2). Overall, our results suggest that L213Q variant can be associated with the lipedema clinical phenotype via a partial loss-of-function mechanism, as the reduction of PGS to hPGS would be slower and less efficient.

## 3. Discussion

While genetic factors apparently regulate subcutaneous adipose tissue distribution, so far, no monogenic cause of nonsyndromic primary lipedema has been discovered [5]. With our study, we argue in favor of the involvement of *AKR1C1* in lipedema. To date, *AKR1C1* has not been implicated in any genetic condition characterized by or including lipedema among its clinical manifestations. The *AKR1C1* variant that we found in this family consisted of a Leu213Gln substitution, located outside the active site of the aldo-keto reductase 1C1 that is predicted to reduce steroid hormones catalysis. Indeed, bioinformatic analysis suggests a partial loss of function of 20α-HSD activity of the mutated AKR1C1. The AKR1C enzymes exert their HSDs activity mostly in subcutaneous adipose tissue as the reduction and inactivation of steroid hormones [27,28]. Specifically, AKR1C1 can catalyze the reduction of progesterone to 20α-hydroxyprogesterone and allopregnanolone to 5α-pregnane-3α-20α diol by its 20α-HSD activity. In this way, AKR1C1 decreases the levels of progesterone and allopregnanolone in peripheral adipose tissue [29]. Interestingly, *AKR1C1* expression was also higher in subcutaneous fat of women with obesity, showing its implication in metabolic disorders [30].

In addition, an *AKR1C1* loss of function could lead to a decrease in progesterone catalysis and a consequent increase of lipogenesis mediated by this steroid hormone. Indeed, previous murine studies have shown that progesterone has lipogenic action on adipose tissue by upregulating adipocyte determination and differentiation 1/sterol regulatory element-binding protein 1c (ADD1/SREBP1c) expression in primary cultured preadipocyte from rat parametrial adipose tissue [31]. ADD1/SREBP1c promotes adipocyte differentiation and gene expression linked to fatty acid metabolism [32]. Transcriptomic and functional analysis of differentiated adipocytes of Landrace piglets showed many significantly enriched lipid deposition and steroid hormone biosynthesis that involved hydroxysteroid dehydrogenases activity of AKR1C1 [33].

Moreover, AKR1C1 is also involved in catalyzing the synthesis of prostaglandins in humans [34]. It has been shown that prostaglandin 2 alpha (PGF2α) inhibited adipogenesis by activating its specific receptor on preadipocytes [35,36]. Since AKR1C1 promotes PGF2α synthesis, its diminished activity in our patients would result in lower levels of PGF2α, and therefore more adipogenesis. However, our bioinformatic analysis focused on AKR1C1 Leu213Gln activity on progesterone, for which quantitative functional data were available, and further computational analyses corroborated by specific functional studies would be needed to assess the consequence of our findings on the interaction with PGF2α.

Finally, although lipedema is typically reported as a painful disorder [37], our patients did not complain of pain or tenderness to palpation. This could be explained by the reduced activity of AKR1C1 on allopregnanolone, a neurosteroid that has an analgesic effect by enhancing GABA_A_ currents [17]. In fact, opening of the GABA_A_ receptor causes depolarization of dorsal root ganglion cells and blocks nociceptive transmission [38]. We speculate that a partial loss of function in 20α-HSD activity of AKR1C1 would result in diminished inactivation of allopregnanolone, that could still exert its analgesic effect.

## 4. Materials and Methods

### 4.1. Ethical Compliance

The study was performed according to the declaration of Helsinki and was approved by thelocal ethics committee (Comitato Etico dell’Azienda Sanitaria dell’Alto Adige, Protocol number 0111181-BZ). Written informed consent was obtained from the family members for publication prior to the study.

### 4.2. Subjects

A 54-year-old female was diagnosed with lipedema at the San Giovanni Battista Hospital, in Rome (Italy). Lipedema occurred symmetrically in the legs and hips. Onset of the condition was around 16 years of age. No obesity-related comorbidities or endocrine alterations were diagnosed. The proband reported that the same condition was also present in her sister and her daughter (Figure 1). Clinical details of affected family members are reported in Table 3.

### 4.3. Whole Exome Sequencing

Genetic testing was performed on germline DNA extracted from the blood of relevant family members. DNA library preparation and exome capture were performed using the Agilent SureSelect Clinical Research Exome kit (54Mb) according to the manufacturer′s protocol. Libraries were pooled post capture. Paired-end sequencing, 2 × 100 bases, was performed on Illumina^®^ HiSeq™4000 platform. PipeMAGI pipeline was used to annotate and filtrate variants as previously described [39]. The American College of Medical Genetics (ACMG) 2015 criteria [40] were used to classify identified variants as pathogenic, likely pathogenic, or variant of uncertain significance (VUS). Variants were also verified on ClinVar (https://www.ncbi.nlm.nih.gov/clinvar/), OMIM (https://www.omim.org/), and VarSome (https://varsome.com/) databases.

### 4.4. Sanger Sequencing Analysis

Identified variants with likely clinical significance were confirmed by Sanger sequencing on a CEQ8800 Sequencer (Beckman Coulter, Brea, CA, USA). The following primers were used to amplify the *AKR1C1* (NM_001353.5) fragment GCTCAAAGAATCTTACCTCATCCC and TATGAATTTCCAGCTCCTTCCAAC, whereas the following internal primer was used to confirm by sequencing the presence of the variant CCTCAACATCACCTGGGATCT.

### 4.5. Quantitative Real-Time Polymerase Chain Reaction

Total RNA was extracted from blood using the Tempus™ Spin RNA Isolation Kit following the manufacturer’s protocol. The SuperScript VILO cDNA Synthesis Kit was used to generate first strand cDNA. Quantitative real-time polymerase chain reaction (qPCR) was performed by using the PowerUp™ SYBR™ Green Master Mix (Thermo Fisher Scientific, Vilnius, Lithuania) on a QuantStudio 3 Real-Time PCR System. The primers used in the qPCR experiments were previously described [41,42] and are the following: GACAAGCTTCCCGTTCTCAG and GGAGTCAACGGATTTGGTCG for *GAPDH*, CCTAAAAGTAAAGCTTTAGAGGCCACC and GAAAATGAATAAGGTAGAGGTCAACATAAT for *AKR1C1*.

### 4.6. Molecular Modeling And Molecular Dynamics (MD) Simulations

MD simulations of human AKR1C1 in ternary complex with NADP+ and 20 α-hydroxyprogesterone (hPGS) were performed using as a starting structure the PDB file with entry 1MRQ [25] by retaining crystallographic water molecules within 10 Å of any atom of either NADP+ or hPGS. In silico mutagenesis of variants E127D, L213Q, H222I, H222S, R304L, Y305F, T307V, and D309L was performed using the highest-ranked non-clashing backbone-dependent rotamer provided by “mutate residue” function of Maestro v. 12.2.012 (Schrodinger LLC, New York, NY, USA) suite. All atom MD simulations were run using GROMACS 2019.2 simulation package [43], adopting CHARMM36m [44] force field. Parameters for hPGS and NADP+ were obtained using the Input Generator module of CHARMM-GUI [45].

All AKR1C1 variants were subjected to the same system preparation and energy minimization as in [46], briefly consisting of placing the protein complex 1.2 nm from dodecahedral simulation box edges. Water molecules were added to the system and neutralized with 150 mM KCl (system size ~47800 atoms), finally subjected to steepest descent (F_max_ = 1000 kJ mol^−1^ nm^−1^) and conjugate gradient (F_max_ = 500 kJ mol^−1^ nm^−1^) minimization of the sidechains. The minimized systems were subjected to the same equilibration procedure as in [47] summarily involving two 2 ns steps at a constant temperature (310 K), with and without position restraints on the backbone atoms. Finally, WILD-TYPE AKR1C1 and L213Q variant underwent extensive 2 × 500 ns MD simulations at constant pressure and temperature (1 atm and 310 K, respectively), whereas all the other variants directly involved in the catalytic process described in [25] were subjected to 20 ns MD simulations due to their impact on the enzyme activity.

The solvent-accessible surface (SAS) and the molecular mechanics interaction energy were calculated using the *gmx sasa* and *gmx energy* functions implemented in GROMACS, respectively. The structural index of flexibility root mean square fluctuation (RMSF), calculated as the time-averaged root mean square deviation of Cα with respect to the average structure, was calculated by GROMACS function *gmx rmsf*. Data reported in Table 1 refer to the average ± standard deviation calculated over 1000 ns MD simulations, whereas data reported in Table 2 refer to the same values calculated only on the first 20 ns of MD simulation of WILD-TYPE and L213Q variant, for timescale consistency with respect to all the other variants. The statistical significance of the differences in SAS and interaction energies observed between WILD-TYPE and L213Q variant was assessed using a one-tailed *t*-test (*p* < 0.001).

The persistence of H-bonds and hydrophobic interactions during the 1000 ns trajectories was calculated using PyInteraph [48], using the parameters for describing sidechain–sidechain interactions defined in [49], and refers to the percentage of frames where the interaction distance and angle constraints were satisfied. This approach has been shown to be able to recognize allosteric properties in protein complexes undergoing extensive MD simulations [47,49].

### 4.7. QSAR Models and Statistical Analysis

A single or double linear correlation analysis was performed between the enzymatic parameters *k_cat_*, *K_M_*, and *k_cat_*/*K_M_* of the reduction of progesterone to 20α-hydroxyprogesterone measured in [25] and the set of descriptors calculated on the MD trajectories by determining the correlation coefficient matrix. The descriptor, or linear combination of descriptors, with the highest correlation coefficient (R^2^) were used for building quantitative structure–activity relationship (QSAR) models. For double linear correlation models, the non-correlation of the independent variables defined by the descriptors was verified by linear regression. The robustness of each of the two final models was assessed using leave-one-out (LOO) cross validation, the statistical quality of the models was verified by the average R^2^ and the mean square error (MSE) descriptor, calculated as follows:(3)$$MSE=\frac{1}{n}\;\overset{n}{\underset{i=1}{\sum }}{\left(x_{i}^{observed}-x_{i}^{predicted}\right)}^{2}$$

## 5. Conclusions

In conclusion, we suggest that regulation of steroid hormone levels by aldo-keto reductase 1C1 plays an important role in the accumulation of subcutaneous adipose tissue. Our results are consistent with *AKR1C1* being the first candidate gene for lipedema.

## Figures and Tables

**Figure 1 ijms-21-06264-f001:**
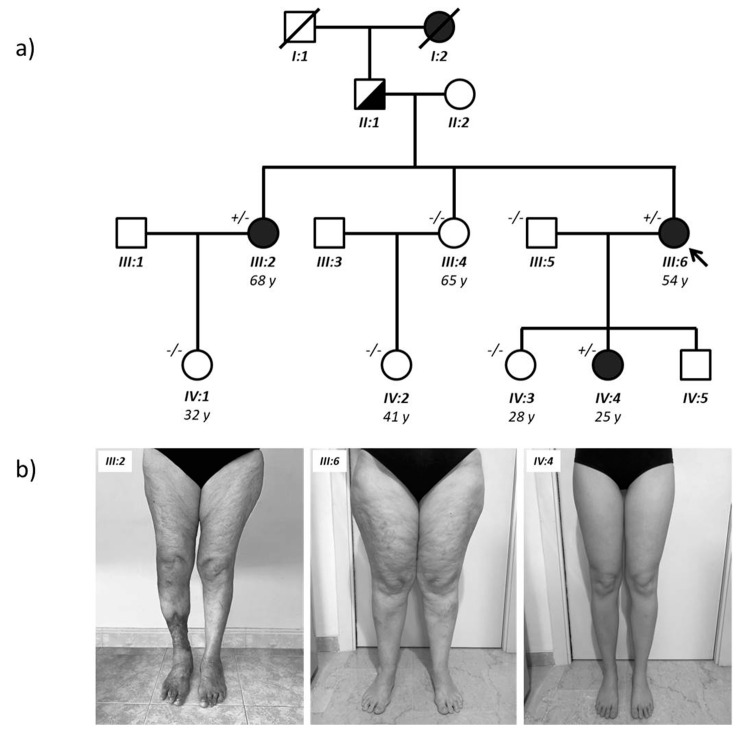
(**a**) Family tree; (**b**) Lipedema of the proband (III:6), her sister (III:2), and her daughter (IV:4). The right leg ulcer of proband’s sister (III:2) is the chronic result of a post traumatic post-phlebitic syndrome. WES analysis was performed on almost all family members of the last two generations and the *AKR1C1* genotype is indicated next to the individual symbols (+/– are affected heterozygotes with the L213Q variant; –/– are unaffected wild-type homozygotes).

**Figure 2 ijms-21-06264-f002:**
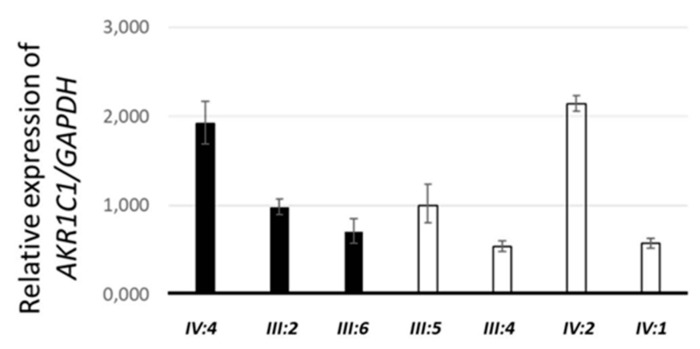
Relative expression of *AKR1C1* in the blood of the affected (black) and healthy (white) family members.

**Figure 3 ijms-21-06264-f003:**
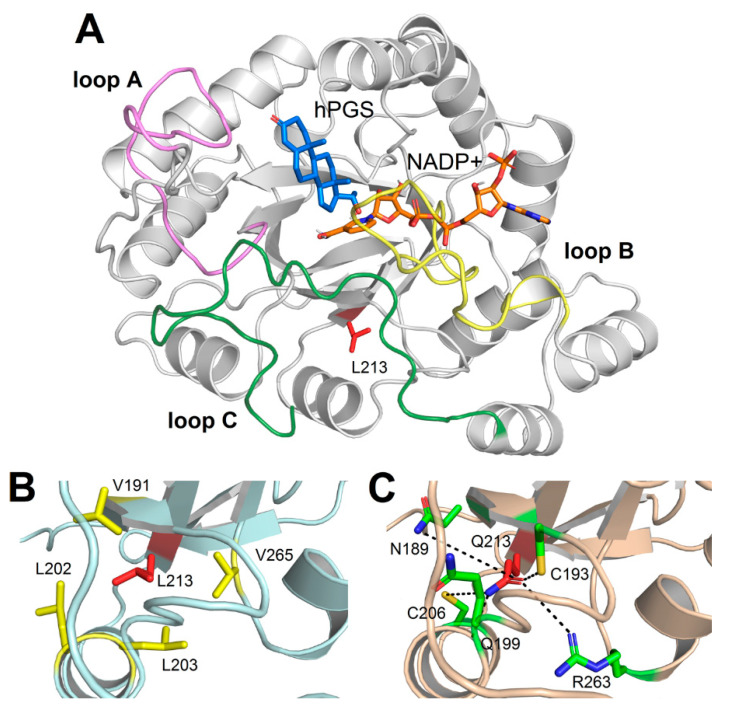
*(***A**) Three-dimensional structure of AKR1C1 complexed with NADP+ and hPGS. Protein structure is shown as grey cartoon, loop A is colored in violet, loop B in yellow, and loop C in green. Residue L213, NADP+, and hPGS are represented as sticks and colored in red, orange, and blue, respectively; O atoms are depicted in red, N atoms in blue, and H atoms in white. For the sake of clarity, only non-polar H atoms are shown; (**B**) Detail of the residues involved in hydrophobic interactions with L213. Protein structure is shown as light blue cartoon; residues are shown as sticks; L213 is shown in red; V191, L202, L203, and V265 are shown in yellow; (**C**) Detail of the residues involved in hydrogen bonds with Q213. Protein structure is shown as light orange cartoon; residues are shown as sticks; Q213 is shown in red; N189, C193, Q199, C206 and R263 are shown in green. O atoms are colored in red, N atoms in blue, and S atoms in yellow, hydrogen bonds are shown as dashed lines. Please note that distances between H-bond donors and acceptors and geometric features are merely representative, as the figure represents a single frame out of 100,000 frames spanned by MD simulations.

**Table 1 ijms-21-06264-t001:** Structural and energetic descriptors calculated over 1000 ns MD simulations. Data are reported as average ± standard deviation, SAS represents the solvent-accessible surface of either NADP+ or hPGS, *IE_P-hP_* is the interaction energy between protein and hPGS, *IE_N-hP_* is the interaction energy between NADP+ and hPGS.

Variant	SAS NADP+ (nm^2^)	SAS hPGS (nm^2^)	*IE_P-hP_* (kJ/mol)	*IE_N-hP_* (kJ/mol)
WILD-TYPE	1.10 ± 0.26	0.98 ± 0.43	–122.91 ± 23.60	–9.75 ± 7.74
L123Q	1.55 ± 0.35	1.46 ± 0.59	–105.66 ± 23.88	–5.22 ± 6.17

**Table 2 ijms-21-06264-t002:** Overview of the MD-derived energetic descriptors and the functional descriptors of AKR1C1 variants. *IE_P-hP_* is the interaction energy between protein and hPGS, *IE_N-hP_* is the interaction energy between NADP+ and hPGS. The novel L213Q variant is highlighted in grey, ↓ and ↑ represent values lower and higher than the WILD-TYPE, respectively.

Variant	*IE_P-hP_* (kJ/mol)	*IE_P-hP_* (kJ/mol)	*K_m_* (μM)	*k_cat_* (min^−1^)	*k_cat_*/*K_m_* (min^−1^ μM^−1^)
R304L	−119.28 ± 81.07	−31.77 ± 8.89	283.3 ± 13.2	96.1	0.3 ↓
E127D	−97.50 ± 63.49	−14.29 ± 9.22	9.4 ± 2.5	4.4	0.4 ↓
H222I	−111.04 ± 67.84	−28.61 ± 8.76	28.6 ± 5.2	18.5	0.6 ↓
H222S	−110.99 ± 68.25	−8.57 ± 5.19	25.6 ± 5.7	39.6	1.5 ↓
L213Q	−114.93 ± 24.89	−8.90 ± 6.87	*5.1*	*38.2*	*7.55 ↓*
T307V	−126.19 ± 70.32	−33.38 ± 7.61	4.1 ± 0.3	41.3	10 ↓
WILD-TYPE	−118.72 ± 13.23	−5.69 ± 7.10	4.2 ± 0.8	65.1	15.2
D309L	−126.98 ± 83.16	−19.78 ± 10.08	4.6 ± 0.6	119.5	25.9 ↑
Y305F	−120.12 ± 84.75	−19.72 ± 6.08	5.6 ± 1.9	156.3	27.9 ↑

**Table 3 ijms-21-06264-t003:** Clinical characteristics of the proband and her family.

Clinical Characteristics	III:2 Sister	III:6 Proband	IV:4 Daughter
Age	68	54	25
Age of onset	Puberty	Puberty	Puberty
Menarche	Regular	Regular	Regular
Height (m)	1.68	1.65	1.75
Weight (Kg)	68	79	56
BMI	24.09 normal	29.02 overweight	18.29 normal
Lipedema stage Localization of fat depots	Stage 2, Type 2 thighs and buttocks	Stage 2, Type 2 thighs and buttocks	Stage 1, Type 2 thighs
Comorbidity (diabetes, hypertension, dyslipidemia)	Hypertension	None	none
Endocrine alteration (e.g., thyroid, insulin-resistance)	Thyroid disfunction	None	none
Pain in the morning (VAS scale 1–10)	0	0	0
Pain at night (VAS scale 1–10)	0	0	0
Anxiety/depression/fatigue	Anxiety	none	none

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
