# Peer review of "Aldo-Keto Reductase 1C1 (AKR1C1) as the First Mutated Gene in a Family with Nonsyndromic Primary Lipedema"

_ijms, 2020, doi:10.3390/ijms21176264_

Round 1

Reviewer 1 Report

The article is very interesting as it discusses the genetics of lipedema. The whole study was based on one family with three affected females. It would be interesting to screen the gene in a larger cohort of families with Lipedema.

Comments:

  1. Introduction: on line 72: The authors mentioned that “in rats progesterone reverses the weight-reducing actions of estradiol. How the authors can relate this to Lipedema?
  2. Subjects: Can the authors mention which stage/type are the affected females?
  3. The sister has thyroid dysfunction. How can the authors interpret this dysfunction in the context of the generated data?
  4. The authors analyzed the sequencing data from DNA extracted from blood. Did the authors try extracting DNA from fat biopsies of affected areas of lipedema females? That would be interesting to see if the data correlates.
  5. It is interesting that the authors included III:5 in figure 2. Can the authors explain why?
  6. On page 7, line 240: the authors mentioned that “the variant does not dramatically affect the protein structure”. Does that mean that the expression of the protein is not altered in Lipedema?  
  7. On page 8, line 290: the authors suggest a partial loss of function in L213Q variant. Can the authors elaborate on this data? How this might affect lipedema patients?
  8. Can the author provide reference to the line 321 on page 9: “PGE2a may exert fat atrophy acting on adipocyte”?
  9. The authors suggest that diminished activity of AKR1C1 can be correlated with deregulation of fat atrophy. But in lipedema we observe hypertrophic adipocytes not fat atrophy. How the authors can explain this?

Author Response

Reviewer 1

The article is very interesting as it discusses the genetics of lipedema. The whole study was based on one family with three affected females. It would be interesting to screen the gene in a larger cohort of families with Lipedema.

Reply: We are glad that this Reviewer found merit in our work. As he/she suggested we are actively screening more lipedema patients in the search of more AKR1C1 mutations. We addressed the other comments as follows:

- Introduction: on line 72: The authors mentioned that “in rats progesterone reverses the weight-reducing actions of estradiol. How the authors can relate this to Lipedema?

Reply: We rephrased this paragraph of the Introduction in order to make more clear that estradiol and progesterone have opposing effects on subcutaneous adipose tissue. As mentioned at the end of the Introduction, AldoKetoReductase (AKR) 1C1 inactivates progesterone, therefore a loss of function variant in AKR1C1 will likely result in more progesterone being locally available in the SC tissue, promoting fat accumulation in adipocytes and eventually Lipedema.

- Subjects: Can the authors mention which stage/type are the affected females?

Reply: We added a line in Table 1 to specify the Lipedema stage in the three affected females.

- The sister has thyroid dysfunction. How can the authors interpret this dysfunction in the context of the generated data?

Reply: We have no particular explanation for this finding in the proband’s sister III:2. She is the only one with thyroid dysfunction: it may be a coincidence.

- The authors analyzed the sequencing data from DNA extracted from blood. Did the authors try extracting DNA from fat biopsies of affected areas of lipedema females? That would be interesting to see if the data correlates.

Reply: We extracted DNA only from blood since this is sufficient to determine the presence of a germinal mutation. Biopsies are not necessary for this purpose, while they are potentially dangerous and not appreciated by the patients themselves. Furthermore, although we cannot formally exclude a double hit mechanism, the AKR1C1 variant of this family appears to work in a dominant fashion and we do not expect to find a second variant in the affected fat tissue.

- It is interesting that the authors included III:5 in figure 2. Can the authors explain why?

Reply: III:5 is not the only spouse included in Figure 1.  All spouses (e.g. III:1, III:3) are included in the family tree. We did make the figure legend more explicit in order to highlight that only females heterozygous for the AKR1C1 variant are affected.  

- On page 7, line 240: the authors mentioned that “the variant does not dramatically affect the protein structure”. Does that mean that the expression of the protein is not altered in Lipedema?

Reply: In the revised manuscript, we now specify that we refer to the observation that the variant does not affect protein three-dimensional structure. Based on this observation we have no reasons to suspect that the point mutation detected in the family may alter the expression of AK1R1C1

- On page 8, line 290: the authors suggest a partial loss of function in L213Q variant. Can the authors elaborate on this data? How this might affect lipedema patients?

Reply: A partial loss of function in the L231Q variant would result in a slower and less efficient reduction of progesterone to hydroxyprogesterone, which in turn could lead to increased subcutaneous fat deposition. This hypothesis has been explicitly stated in the manuscript, and in the absence of further lines of evidence we prefer to limit our speculations as to other possible implications.

- Can the author provide reference to the line 321 on page 9: “PGE2a may exert fat atrophy acting on adipocyte”?

Reply: We added one relevant reference providing strong evidence for this effect [Taketani et al. 2014 doi: 10.1167/iovs.13-12589] and rephrased this paragraph of the Discussion in order to better explain that PGE2a actually inhibits adipogenesis by acting on its specific receptor.

- The authors suggest that diminished activity of AKR1C1 can be correlated with deregulation of fat atrophy. But in lipedema we observe hypertrophic adipocytes not fat atrophy. How the authors can explain this?

Reply: As stated above, we rephrased the paragraph of the Discussion that possibly confused the Reviewer. Obviously in lipedema there is no fat atrophy but just the opposite i.e. hypertrophic adipocytes and we propose that this results from two concomitant effects of AKR1C1 partial loss-of-function: 1) diminished inactivation of progesterone (i.e. more progesterone available which has an adipogenic effect), and 2) diminished synthesis of PGE2a (i.e. less PGE2a which has an antiadipogenic effect).  We hope that the revised Discussion is now more clear.

Reviewer 2 Report

The manuscript by Michelini et al., reporting AKR1C1 as the first candidate gene associated with non-syndromic lipedema is a very straight forward work and appears to be carefully performed. The authors have been very resourceful in WES analysis. The methodology is appropriate and the results are interesting and were presented smoothly. Overall, the article is well presented. However, a minor revision is needed:

1. A table or pie chart illustrating the Number of effects by functional class (count and percentage for missense, nonsense and silent mutations) and Number of effects by type and region

2. In fig 3 b and c, the difference in the ptn structure is not clear. It would be better to display the superimposed image of WT & variant overlap and the changes in amino acids (L213Q) can be cleary shown (this can be easily implemented by superPose at http://superpose.wishartlab.com/)

I recommend this publication with the minor revisions stated.

Author Response

Reviewer 2

The manuscript by Michelini et al., reporting AKR1C1 as the first candidate gene associated with non-syndromic lipedema is a very straight forward work and appears to be carefully performed. The authors have been very resourceful in WES analysis. The methodology is appropriate and the results are interesting and were presented smoothly. Overall, the article is well presented. However, a minor revision is needed:

Reply: We are glad that this Reviewer found merit in our work. We have addressed all the points raised as follows.

  1. A table or pie chart illustrating the Number of effects by functional class (count and percentage for missense, nonsense and silent mutations) and Number of effects by type and region

Reply: The Reviewer is probably suggesting to prepare a pie chart summarizing the pathogenic AKR1C1 variants divided by functional class (missense, nonsense, frameshift, splicing, etc.) that would put the Leu213Gln variant of this family into perspective. However, this is not possible because this is the FIRST pathogenic variant described for the AKR1C1 gene (see OMIM entry *600449 – https://omim.org/entry/600449) and there are no other pathogenic variants to compare.

  1. In fig 3 b and c, the difference in the ptn structure is not clear. It would be better to display the superimposed image of WT & variant overlap and the changes in amino acids (L213Q) can be clearly shown (this can be easily implemented by superPose at http://superpose.wishartlab.com/)

Reply: We agree with the Reviewer that panel b and c of Figure 3 could be improved for better clarity. Therefore we prepared a new version of Figure 3 where the pattern of hydrogen-bonds involving the mutated side chains are highlighted. The figure legend was modified accordingly.

We also tried to superpose the backbone structure of the WT and the variant, as suggested by the Reviewer, however the resulting figure (see the attached file) is not very informative, as structural differences are indeed minor, but they are  appreciable in the time-frame spanned by our molecular dynamics simulations (as detailed in Table 2).

I recommend this publication with the minor revisions stated.

Reviewer 3 Report

In the manuscript (ID: ijms-889202), the authors of Michelini et al present studies on the discovery of a genetic variant (p.Leu213Gln) in AKR1C1, a gene encoding aldo / keto reductase catalyzing the reduction of progesterone to its inactive form, 20-α-hydroxy-progesterone. The authors conducted a comparative molecular dynamics simulations of the wild type enzyme and the variant and QSAR analysis in order to explain the impact of this variant on the enzyme partial loss-of-function in the case of lipedema.

The manuscript is very interesting, data well-presented and quite well written.

I have a few minor concerns that can be easily addressed:

1) Introduction, line 91, abbreviation “AKR1C1” should be explained,

2) In the section “2.6. Molecular modelling and Molecular dynamics (MD) simulations”, lines 145-149, the methodology should be described in detail not by using reference to the articles that creates confusion in understanding the methodology step by step,  

3) In line 214, abbreviation “TIM ” should be explained,

4) The units in Table 2 for SAS are not the same as in the text in line 251-252 and should be corrected,

5) The abbreviation “SAS” in Table 2 is not explained.

Author Response

Reviewer 3

In the manuscript (ID: ijms-889202), the authors of Michelini et al present studies on the discovery of a genetic variant (p.Leu213Gln) in AKR1C1, a gene encoding aldo / keto reductase catalyzing the reduction of progesterone to its inactive form, 20-α-hydroxy-progesterone. The authors conducted a comparative molecular dynamics simulations of the wild type enzyme and the variant and QSAR analysis in order to explain the impact of this variant on the enzyme partial loss-of-function in the case of lipedema.

The manuscript is very interesting, data well-presented and quite well written. I have a few minor concerns that can be easily addressed.

Reply: We are glad that this Reviewer found merit in our work. We have addressed all the points raised as follows.

1) Introduction, line 91, abbreviation “AKR1C1” should be explained.

Reply: This was done in the revised manuscript.

2) In the section “2.6. Molecular modelling and Molecular dynamics (MD) simulations”, lines 145-149, the methodology should be described in detail not by using reference to the articles that creates confusion in understanding the methodology step by step.

Reply: As requested, we have now explained the computational protocols in greater detail.

3) In line 214, abbreviation “TIM ” should be explained.

Reply: The acronym TIM has been explained in the revised manuscript.

4) The units in Table 2 for SAS are not the same as in the text in line 251-252 and should be corrected.

Reply: We apologize for the typos. Units are now consistent with each other.

5) The abbreviation “SAS” in Table 2 is not explained.

Reply: It is now explained.